

# INFERNO: a fire and emissions scheme for the Met
# Office's Unified Model
Stephane Mangeon[1,2], Apostolos Voulgarakis[1], Richard Gilham[2], Anna Harper[3],
Stephen Sitch[4], Gerd Folberth[2]
[1]Department of Physics, Imperial College London, London, United Kingdom
[2]Met Office, FitzRoy Road, Exeter, EX1 3PB, UK
[3]College of Engineering, Mathematics, and Physical Sciences, University of Exeter, Exeter, UK
[4]College of Life and Environmental Sciences, University of Exeter, Exeter, UK
*Correspondence to*: Stéphane Mangeon (stephane.mangeon12@imperial.ac.uk)
**Abstract.** Warm and dry climatological conditions favour the occurrence of forest fires. These fires then
become a significant emission source to the atmosphere. Despite this global importance, fires are a local
phenomenon and are difficult to represent in a large-scale Earth System Model (ESM). To address this,
the INteractive Fire and Emission algoRithm for Natural envirOnments (INFERNO) was developed.
INFERNO follows a reduced complexity approach and is intended for decadal to centennial scale climate
simulations and assessment models for policy making. Fuel flammability is simulated using temperature,
relative humidity, fuel density as well as precipitation and soil moisture. Combining flammability with
ignitions and vegetation, burnt area is diagnosed. Emissions of carbon and key species are estimated
using the carbon scheme in the JULES land surface model. JULES also possesses fire index diagnostics
which we document and compare with our fire scheme. Two meteorology datasets and three ignition
modes are used to validate the model. INFERNO is shown to effectively diagnose global fire occurrence
(R=0.66) and emissions (R=0.59) through an approach appropriate to the complexity of an ESM,
although regional biases remain.





## 1 Introduction

Fire is a key interaction between the atmosphere and the land surface (Bowman et al., 2009). Its impacts are wide-ranging: it influences forest succession (Bond and Keeley, 2005), is a tool for deforestation (van der Werf et al., 2009) and is an important natural carbon source (Bowman et al., 2013), while it also provides a major natural hazard to humans through property and infrastructure destruction and air quality degradation (Johnston et al., 2012; Marlier et al., 2013). Not only are biomass burning emissions substantial in magnitude (Lamarque et al., 2010), they also drive the variability of atmospheric composition (Spracklen et al., 2007; Voulgarakis et al., 2010, 2015) and impact short-term climate forcing (Tosca et al., 2013).

There are feedbacks between fire and climate: low-humidity conditions cause droughts, which enhance fire activity (Field et al., 2009), which, in turn, emits aerosols and trace gases (Akagi et al., 2011), influencing the abundances of radiatively active atmospheric constituents, cloud formation and lifetime, and in turn precipitation, and surface albedo (Voulgarakis and Field, 2015). Bistinas et al. (2014) showed global fire frequency is correlated with land-use, vegetation type and meteorological factors (dry days, soil moisture and maximum temperature) and human presence tends to noticeably reduce fire activity (land-management, landscape fragmentation and urbanization). Examining and quantifying such impacts and feedbacks is paramount to Earth System Models (ESMs), yet to integrate vegetation fires presents many challenges as it intricately links multiple disciplines from ecology to atmospheric chemistry and physics and climate science.

Integration of fires into Dynamic Global Vegetation Models (DGVMs) was the first step towards fire within ESMs (e.g. (Arora and Boer, 2005; Fosberg et al., 1999; Li et al., 2012; Pfeiffer et al., 2013; Sitch et al., 2003; Thonicke et al., 2001, 2010; Venevsky et al., 2002; Yue et al., 2014). Vegetation fires have been implemented into only a few ESMs, e.g. ECHAM (Lasslop et al., 2014) and the Community ESM (Li et al., 2013, 2014, p.2).

Here, we present and evaluate the INteractive Fire and Emission algoRithm for Natural envirOnments (INFERNO) and its implementation. INFERNO is a necessarily simple parameterization that focuses on the large-scale occurrence of fires and is suitable for ESM application. The model uses a few key driving variables while retaining a broadly accurate parameterization for fire emissions. INFERNO's performance against observations and well established and operationally relevant fire indices is presented.

## 2 Model description

### 2.1 INFERNO

INFERNO was constructed upon the simplified parameterization for fire counts proposed and evaluated for the present-day by (Pechony and Shindell, 2009), which was subsequently shown to provide a good estimate for large-scale fire variability over climatological timescales (Pechony and Shindell, 2010). In short, that parameterization used monthly mean temperature, relative humidity and precipitation to simulate fuel flammability. It also used human population density and lightning to represent ignitions. To incorporate this parameterization within the Joint UK Land Environment Simulator (JULES, Best et



al., 2011; Clark et al., 2011), several changes were applied. Upper layer soil moisture is used to represent
precipitation memory while precipitation acts as a rapid fire deterrent. Vegetation Density was replaced
by Fuel Density, an index dependent on leaf carbon and Decomposable Plant Material (DPM), i.e. litter.
Such a relationship with fine fuel and moisture was used in Thonicke et al. (2001). Furthermore, we
developed a parameterization to obtain burnt area (BA), emitted carbon (EC) and fire emissions of
different species ($E_X$) and our fire diagnostics are made for each of the nine Plant Functional Types
(PFTs) in the current version of JULES (Harper et al., submitted).
Figure 1 summarizes the mechanisms of INFERNO, and Fig. A1 illustrates the dependence of INFERNO
on individual driving variables.

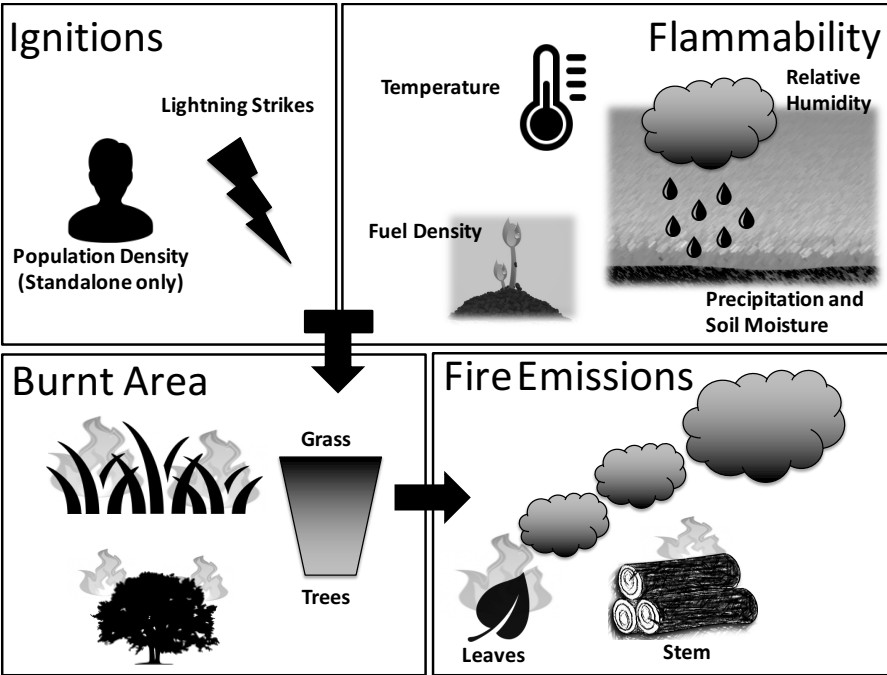


**Fig. 1. Schematic summarizing the INteractive Fire and Emission algoRithm for Natural envirOnments**
**(INFERNO) and its key components and behaviour. Ignitions can be accounted for in a variety of ways (see**
**Sect. 2.1.1), meteorology influences flammability (see Sect. 2.1.2), while plant coverage influences burnt area**
**(see Sect. 2.1.3), finally emissions are calculated according to leaf and stem carbon for each PFT (see Sect.**
**2.1.4).**
**2.1.1 Ignitions ($I$)**
INFERNO calculates ignitions in either one of three modes:
First, we can assume constant or ubiquitous ignitions, currently calibrated to a global average of $I_T =$
1.67 ignitions km$^{-2}$ month$^{-1}$. This corresponds to 1.5 ignitions km$^{-2}$ month$^{-1}$ due to humans ($I_A$),
heuristically determined, and 0.17 ignitions km$^{-2}$ month$^{-1}$ natural ignitions due to lightning ($I_N$), derived
from the multi-year annual mean of 2.7 strikes km$^{-2}$ year$^{-1}$ (Huntrieser et al., 2007) assuming 75% of
strikes being cloud-to-ground (Prentice and Mackerras, 1977). This mode inherently suppresses the
variability in fires due to any anthropogenic or natural ignition changes (Pechony and Shindell, 2009,

85    2010).



Second, human ignitions and suppressions can be assumed to remain constant at the global mean value
mentioned above ($I_A$ = 1.5 ignitions km$^{-2}$ month$^{-1}$), however cloud-to-ground lightning strikes may vary,
and in addition each strike is assumed to start a fire. This mode accounts for natural variability in fire
ignitions, which can be simulated within an ESM, or prescribed from observations.
Third, varying human ignitions and suppressions and varying natural ignitions (cloud-to-ground
lightning strikes, as in mode 2). This was the original ignition approach in Pechony and Shindell (2009),
which was left unchanged and is detailed below. In this ignition mode, anthropogenic ignition and
suppression depends on population density ($PD$), as proposed by Venevsky et al. (2002).
$I_A = k(PD) \, PD \, \alpha$         (1)
$PD$ is in units of people km$^{-2}$, and $k(PD) = 6.8 \times PD^{-0.6}$ is a function that represents the varying
anthropogenic influence on ignitions in rural versus urban environments. The parameter $\alpha = 0.03$
represents the number of potential ignition sources per person per month per km$^2$. Both natural and
anthropogenic ignitions have the potential to be suppressed by humans, such that the fraction of fires not
suppressed is:
$f_{NS} = 7.7 \, (0.05 + 0.9 \times e^{-0.05 \, PD})$         (2)
Equation 2 includes a scaling factor (Pechony and Shindell, 2009) originally introduced to calibrate the
number of fires to MODIS observations. Assuming no suppression for the first two ignition modes
($f_{NS} = 1$), total ignitions ($I_T$, in units, ignitions m$^{-2}$ s$^{-1}$) can be represented as (Eq. 3):
$I_T = (I_N + I_A) \, f_{NS} / (8.64 \times 10^{10})$         (3)
Dividing by $8.64 \times 10^{10}$ converts ignitions km$^{-2}$ month$^{-1}$ to ignitions m$^{-2}$ s$^{-1}$.
**2.1.2 Flammability ($F$)**
We adapt the (Pechony and Shindell, 2009) scheme for flammability to function interactively within an
ESM (see Eq. 6). Starting from the saturation vapour pressure ($e^*$, Eq. 4; Goff and Gratch, 1946) and
its temperature dependence, we introduce a Fuel Density index ($FD_{PFT}$, Eq. 5) as well as Relative
Humidity ($RH$), precipitation and soil moisture in order to obtain Flammability (Eq. 6). The land surface
model (JULES) determines soil moisture content ($\theta$) and fuel density ($FD$).
$\log_{10}(e^*) = a\left(\frac{T_s}{T}-1\right) + b \log_{10}\left(\frac{T_s}{T}\right) + c\left(10^{d\left(1-\frac{T_s}{T}\right)}-1\right) + f\left(10^{h\left(\frac{T_s}{T}-1\right)}-1\right)$     (4)
As illustrated in Eq. 4, INFERNO utilizes temperature ($T$ in K, at 1.5 m height). The Goff-Gratch (Eq.
4) uses the constants: $a = -7.90298$; $b = 5.02808$; $c = -1.3816 * 10^{-7}$; $d = 11.344$; $f = 8.1328 *$
$10^{-3}$; $h = -3.49149$ and the water boiling point temperature $T_s = 373.16$ K.
$FD_{PFT} = \begin{cases} 1 \text{ for } Fuel_{high} < (DPM_C + Leaf_{C,PFT}) \\ \frac{(DPM_C + Leaf_{C,PFT})}{Fuel_{high} - Fuel_{low}} \text{ for } Fuel_{low} < (DPM_C + Leaf_{C,PFT}) < Fuel_{high} \\ 0 \text{ for } Fuel_{low} > (DPM_C + Leaf_{C,PFT}) \end{cases}$     (5)
Equation 5 shows $FD$ is taken as the PFT-specific leaf carbon ($Leaf_{C,PFT}$) plus the carbon within
decomposable plant material ($DPM_C$). DPM is a soil carbon pool of which we assume 70% is available
to fires i.e. near-surface (DPM is shared across all PFTs). $FD$ scales linearly between 0 (at a threshold
of $Fuel_{low} = 0.02$ kgC m$^{-2}$) and 1 (at a threshold of $Fuel_{high} = 0.2$ kgC m$^{-2}$). Similar approaches to





represent fuel availability within fire parameterizations have commonly been adopted (Arora and Boer,
2005; Li et al., 2012; Thonicke et al., 2010).
$$F_{PFT} = e^* \left(RH_{up} - RH\right)/(RH_{up} - RH_{low}) \, e^{-2R} \, FD_{PFT} \, (1 - \theta) \tag{6}$$
$RH$ is the relative humidity (%) and $R$ is the precipitation rate (mm day$^{-1}$). The influence of relative
humidity ($RH$) scales between (and is bound by): 0 (at a threshold of $RH_{low} = 10\%$) and 1 (at a threshold
of $RH_{up} = 90\%$). We then adapt the formula by replacing a vegetation index dependent on leaf area
index with the Fuel Density index (FD). Finally, Flammability ($F_{PFT}$) is dependent on upper-level (down
to 0.1 m) soil moisture: $\theta$ is the unfrozen soil moisture as a fraction of saturation. The individual
importance of these variables to our model is illustrated in Fig. A1.

**2.1.3 Burnt Area (BA)**

Our approach is to associate an average burnt area per fire to each PFT, effectively decoupling the fire-
spread stage from local meteorology and topography, which is typically not resolved in the relatively
coarse grid of an ESM. An average burnt area ($\overline{BA_{PFT}}$) was heuristically determined for each PFT: 0.6,
1.4 and 1.2 km$^2$ for trees, grass and shrubs, respectively, such that grass and shrubs will fuel larger fires
than trees. Observational evidence supports that the land cover type is an efficient way to characterize
fires, which tend to be larger in grasslands than in forests (Chuvieco et al., 2008; Giglio et al., 2013).
The $BA$ is then calculated following Eq. 7:
$$BA_{PFT} = I_T F_{PFT} \overline{BA_{PFT}} \tag{7}$$
Here $BA_{PFT}$ is the burnt area (fraction of PFT cover burnt per second) for each PFT; meanwhile the
number of ignitions times the flammability ($I_T F_{PFT}$) represents the number of fires.
Inferring burnt area from number of fires in this manner stands out from other fire models which utilize
wind speed (Arora and Boer, 2005; Thonicke et al., 2010; Li et al., 2012), effectively modelling the fire
rate of spread. Wind is key to the modelling of individual fires; yet implementing wind effectively within
fire models designed for the relatively coarse grid of ESMs was found to be problematic (Lasslop et al.,
2014, 2015). Conversely, Hantson et al. (2014) found global fire size was mostly influenced by
precipitation, aridity and human activity (population density and croplands).

**2.1.4 Emitted Carbon (EC)**

To account for the wetness of fuel in INFERNO, combustion completeness (the fraction of biomass
exposed to a fire that was volatized) scales linearly with soil moisture (as a fraction of saturation) with
different upper and lower boundaries for leaf and stem carbon.
$$EC_{PFT} = BA_{PFT} \sum_{leaf,stem}^{i} \left(CC_{min,i} + \left(CC_{max,i} - CC_{min,i}\right)(1 - \theta)\right) C_i \tag{8}$$
Equation 8 shows how the PFT-specific emitted carbon (EC, in kgC m$^{-2}$ s$^{-1}$) is computed. BA is the burnt
area (fraction s$^{-1}$), $CC_{min}$ and $CC_{max}$ are the minimum and maximum combustion completeness for both
leaves ($CC_{min} = 0.8$ and $CC_{max} = 1.0$) and stems ($CC_{min} = 0.8$ and $CC_{max} = 1.0$), $C_i$ is the carbon
stored in each PFT's leaves or stems (kgC m$^{-2}$). The parameters used for combustion completeness
($CC_{min}$ and $CC_{max}$) are similar to the Global Fire Emission Database (GFED(van der Werf et al., 2010),
albeit with lower minimum combustion of stems (0.0 as opposed to 0.2). This change is justifiable by





the difference in the moisture used. Indeed GFED uses a more complex representation of moisture across
multiple fuel types, while our scheme only relies on soil moisture.
**2.1.5 Emitted Species ($E_X$)**
There has been a significant amount of work on estimating emission factors (EFs) across fire biomes
(such as savannahs, boreal forest etc.). This was synthesized in Akagi et al. (2011) as well as Andreae
and Merlet (2001) and its updates. To convert these biome-specific EFs to PFT specific EFs, each PFT
was linked to a fire biome (see Table A1). INFERNO uses these to estimate emissions (Eq. 9).
$E_{X,PFT} = EC_{PFT} \ EF_{X,PFT} \ /[C]$             (9)
Here $E_X$ is the amount of species X emitted by fires (in kg m$^{-2}$ s$^{-1}$), $EC$ is the emitted carbon (in kgC m$^{-2}$
s$^{-1}$) and $EF_X$ is the PFT-specific emission factor (see Table 1) (in kg of species emitted per kg of biomass
burnt), and $[C]$ is the dry biomass carbon content, express as a percentage (Lamlom and Savidge, 2003).
INFERNO currently provides emissions for basic trace gases: $CO_2$, CO, $CH_4$, NO$_x$, $SO_2$ and aerosols:
organic carbon (OC) and black carbon (BC).
**Table 1. INFERNO's emission factors per PFT created from the emission profiles in Akagi et al. (2011), such**
**that each PFT was attributed a fire biome (see Suppl. 2). This method of attributing emission factors to PFTs**
**is similar to that presented in Thonicke et al. (2010), and can be extended to include all species of trace gases**
**and aerosols compiled in Akagi et al. (2011).**

| Emission Factors (g / kg) | CO$_2$ | CO | CH$_4$ | NO$_x$ | SO$_2$ | OC | BC |
|---|---|---|---|---|---|---|---|
| Broadleaf Evergreen Tree (Tropical) | 1643 | 93 | 5.07 | 2.55 | 0.40 | 4.71 | 0.52 |
| Broadleaf Evergreen Tree (Temperate) | 1637 | 89 | 3.92 | 2.51 | 0.40* | 8.2** | 0.56** |
| Broadleaf Deciduous Tree | 1643 | 93 | 5.07 | 2.55 | 0.40 | 4.71 | 0.52 |
| Needleleaf Evergreen Tree | 1637 | 89 | 3.92 | 2.51 | 0.40* | 8.2** | 0.56** |
| Needleleaf Deciduous Tree | 1489 | 127 | 5.96 | 0.90 | 0.40* | 8.2** | 0.56** |
| C3 grass | 1637 | 89 | 3.92 | 2.51 | 0.40* | 8.2** | 0.56** |
| C4 grass | 1686 | 63 | 1.94 | 3.9 | 0.48 | 2.62 | 0.37 |
| Evergreen Shrub | 1637 | 89 | 3.92 | 2.51 | 0.40* | 8.2** | 0.56** |
| Deciduous Shrub | 1489 | 127 | 5.96 | 0.90 | 0.40* | 8.2** | 0.56** |

*Profile not available in Akagi et al. (2011), therefore we mimic tropical forests; **from Andreae and Merlet (2001).
**2.2 Implementation within JULES**
INFERNO is currently implemented within the Joint UK Land Environment Simulator (JULES). (Best
et al., 2011; Clark et al., 2011) its carbon fluxes and vegetation dynamics. The results shown here used
JULES v4.3.1 and INFERNO will be included in JULES from version 4.5 onwards. INFERNO utilizes
soil moisture (see Eq. 6,8) which JULES calculates as the balance between precipitation (following the





scheme for rainfall interception in (Johannes Dolman and Gregory, 1992)) and extraction by
evapotranspiration and runoff (Cox et al. 1999; Best et al. 2011). JULES has four soil layer, and
INFERNO uses the top layer unfrozen soil moisture (0 to 0.1 m depth). Note that in its current state,
JULES does not associate carbon pools with depths, hence it is not possible to access the top-most DPM
only for example. The vegetation dynamics and litter carbon used obey the TRIFFID DGVM (Cox,

186    2001).

In JULES, vegetation carbon content is determined by the balance between photosynthesis, respiration,
and litterfall. Within JULES, TRIFFID (the Top-down Representation of Foliage and Flora Including
Dynamics; Cox et al., 2001) predicts changes in biomass and the fractional coverage of nine plant
functional types (Table A1) based on accumulated carbon fluxes and height-based competition, where
the tallest trees have the first access to space (Harper et al. *In Prep*). Vegetation can grow in height, and
the carbon in leaves, roots, and wood is related allometrically to the "balanced LAI", $L_b$ (Cox et al. 2001).
$L_b$ is the seasonal maximum leaf area index (LAI) and a function of plant height. Within INFERNO, leaf
carbon ($Leaf_C$, used for calculating FD and emissions) is:
$Leaf_C = \sigma_l L_b$            (10)
Meanwhile, wood carbon ($Wood_C$, which affects emissions), is calculated as:
$Wood_C = a_{wl} L_b^{b_{wl}}$            (11)
PFT dependent parameters($\sigma_l$, the Specific Leaf Density, $a_{wl}$, the allometric coefficient and $b_{wl}$, the
allometric exponent) are given in Table A1.
When using JULES in its standalone version, INFERNO can use inputs of population density (in people
km$^{-2}$) and cloud-to-ground lightning flash rates (in flashes km$^{-2}$ month$^{-1}$) from ancillary datasets.
Similarly, meteorology needs to be prescribed and is then interpolated from its native temporal resolution
to the model's time-step. Although designed to be integrated within an ESM, the capability to run
INFERNO with JULES only is particularly useful for present-day comparison with observations, and to
dissociate causes of biases in results.

## 2.3 Fire Weather Indices

Three other well-established daily fire indices are also available within JULES. These indices have been
used for several decades to help plan operational response to wildfires on Numerical Weather Predictions
(NWP) timescales. Although unit-less and ill-defined risk-based quantities, comparison to INFERNO is
still useful for understanding the results in the context of practically established metrics.
The Canadian Fire Weather Index (Forestry Canada, 1992; Van Wagner and Pickett, 1985) consists of
six components, calculated from basic meteorological parameters. Three are fuel moisture codes
designed to represent the drying of different fuel types, their characteristics are displayed in Table A2.
Two intermediate quantities, the Initial Spread Index and the build-up index are calculated from these,
and are in turn used to yield the final Fire Weather Index.
The McArthur Forest Fire Danger Index (Noble et al., 1980; Sirakoff, 1985) was developed for use in
Australia. Simpler in its formulation than the Canadian index, it consists of a drought component
modified by the local temperature, humidity and wind speed. The calculation of the drought component

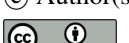



depends on the soil moisture deficit (the amount of water needed to restore the soil moisture content of
the top 800 mm of soil to 200 mm), which is related to the JULES soil moisture.
Finally, the Nesterov Index (Nesterov, 1949) is the simplest fire index implemented in JULES. It uses
only the daily mean temperature, mean daily dew point (or suitable substitute), daily total precipitation
and the previous day's index. The index is incremented daily, unless daily precipitation exceeds 3 mm,
in which case it is reset. The Nesterov index is a key component for other fire models (Venevsky et al.,
2002; Thonicke et al., 2010).
**3 Model configuration**
Monthly lightning data was obtained from LIS-OTD (Lightning Imaging Sensor-Optical Transient
Detector) observations for 2013 (Christian et al., 2003) and was recycled for every year in the simulation.
These detections were converted to cloud-to-ground strikes using the relationship presented in (Prentice
and Mackerras, 1977). Land use and population density were obtained from the HYDE dataset (Hurtt et
al., 2011) and then linearly interpolated to create inter-annually varying data. Finally annual $CO_2$
concentrations, which affect vegetation dynamics, were prescribed as a global average following the
dataset prepared for the global carbon budget (Le Quéré et al., 2015).
To test the sensitivity to the meteorological input, JULES simulations were driven by meteorology from
both CRU-NCEP (Climate Research Unit and -National Center for Environmental Prediction) v5
(http://dods.extra.cea.fr/data/p529viov/cruncep/), and WFDEI (Weedon et al., 2014) with precipitation
from the GPCC (Schneider et al., 2013). Both datasets were used on a 6-hourly basis.
Outside of these driving variables, JULES was configured according to the TRENDY project (Sitch et
al., 2015)(Peng et al., 2015)(Peng et al., 2015). 100 year spin-up was performed repeating the 1990-2000
conditions tenfold. Four configurations were used to create simulations covering 1990-2013, although to
validate INFERNO only the 1997-2010 period was analysed. The first three use CRU-NCEP
meteorology with each of our three ignitions modes (see Sect. 2.1.1); constant ignitions (mode 1),
prescribed lightning and constant anthropogenic ignitions (mode 2), and both natural and anthropogenic
ignitions varying with prescribed lightning and population density (mode 3). The fourth simulation
assumes mode 1 (constant ignitions), while meteorology is prescribed from WFDEI and precipitation
from GPCC.
**4 Results**
Maps of the burnt area and emitted carbon are displayed in Fig. 2, their resolution is 192 longitudes by
145 latitudes grid-cells ($1.875^o$x$1.24^o$). The results from INFERNO used a configuration with CRUNCEP
meteorology and the third ignition mode: interactive lighting and anthropogenic ignitions. We compare
our results with downscaled means from GFED. Note GFEDv4s' burnt area (http://globalfiredata.org,
manuscript in preparation) differs from GFEDv4's (Giglio et al., 2013) as it includes small fires
(Randerson et al., 2012). Over the total study period, INFERNO diagnoses accurate global fire
occurrence and emissions (with R=0.66 for burnt area and R=0.59 for emitted carbon). In addition,
regional mean yearly budgets are compared with GFED in Table B1. We notice burnt area is higher in





all regions other than Australia and New Zealand, and southern hemisphere Africa. Meanwhile emitted
carbon is underestimated in boreal regions and equatorial Asia, but overestimated in most other regions
(significantly in southern hemisphere America). GFEDv4 observes the grid-box with maximum burnt
area within the Central African Republic (87% of grid fraction burnt per year), while INFERNO finds a
maximum burnt area of 57%, slightly to the North (south-east of lake Tchad). The discrepancy is much
larger for emissions, with a maximum emitted carbon of 1.47 kg per $m^2$ in Indonesia predicted by
GFEDv4s, against 0.4 kg per $m^2$ for INFERNO, in Angola. These results could be expected, as
INFERNO focuses on capturing global biomass burning, it will not represent such extremes of burning,
furthermore the immense emitted carbon observed in Indonesia follows from undiagnosed peat fires.

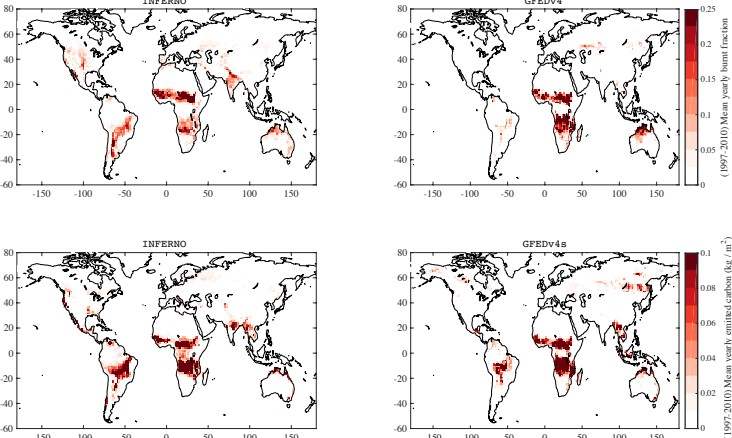


**Fig. 2. 1997-2010 mean yearly burnt fraction (above) and emitted carbon (below, in kg m$^{-2}$). Shown for**
**INFERNO on the left (with CRUNCEP meteorology and interactive ignitions: mode 3) and for GFED on the**
**right.**
Figure 3 shows the modelled global annual average biomass burning emissions and burnt area from 1997
to 2010. The three ignition methods are evaluated: fully interactive ignitions (red) predict the highest
carbon emissions while interactive lightning with constant human ignitions (blue) the lowest. WFDEI
was observed to lead to more biomass burning emissions in tropical forest areas (and in particular the
borders of rainforests), while CRU-NCEP favoured burning in near-desert areas (the Sahel, India and
south American grasslands). We expect this result to be significantly influenced by differences in
precipitation (GPCC for WFDEI runs and CRU for CRU-NCEP; (Schneider et al., 2013).



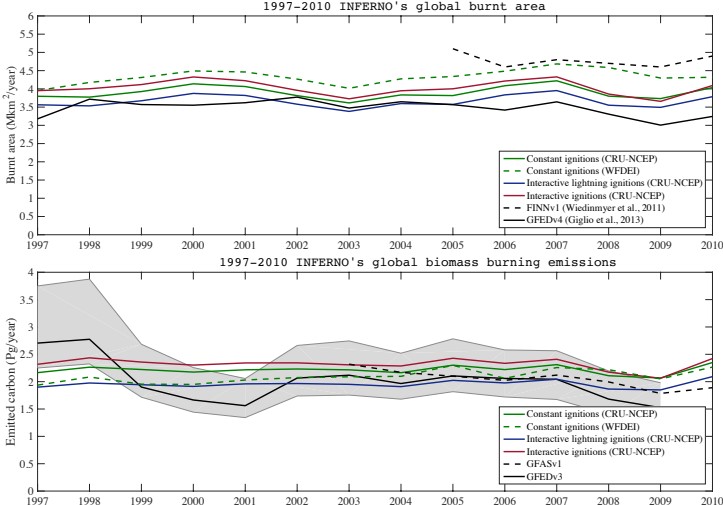


**Fig. 3. 1997-2010 biomass burning emissions and burnt area predicted by INFERNO. Two driving datasets**
**were used, CRU-NCEP (solid lines) and WFDEI (green dotted line). Observations are shown in black**
**(MODIS-based estimates).**

Comparisons with GFASv1 ( and GFEDv3 for emissions (the grey shading represents one standard
deviation within GFEDv3's estimates), to FINNv1 and GFEDv4 for burnt area, were restricted to their
budgets published in (Kaiser et al., 2012; van der Werf et al., 2010; Wiedinmyer et al., 2011; Giglio et
al., 2013) respectively. We also calculated global emissions from GFEDv4s (http://globalfiredata.org,
manuscript in preparation), which adds a small fire contribution (Randerson et al., 2012) to GFEDv4's
burnt area.
Biomass burning emissions and burnt area simulated by the model follow similar trends to GFEDv3,
although with a smaller inter-annual variability in the model. Carbon emissions from all simulations fall
within one standard deviation of GFEDv3, apart from three years: 1997, 1998 and 2001. Note that for
these years, emissions in GFED were obtained from the lower resolution AVHRR rather than MODIS.
1997 and 1998 were strong El-Niño years during which droughts in equatorial Asia led to extreme
emissions from land-clearing fires, a recurrent problem in the region (Field et al., 2009). Indeed in 1997,
in the region contained between 20S-20N and 90E-160E (or equatorial Asia), GFEDv3 estimate
emissions of 1.07 PgC, while INFERNO (with CRU-NCEP and fully interactive ignitions) estimates
0.15 PgC. Unfortunately, peat is not modelled in JULES and thus neither is peat present in our fire
scheme. It was estimated tropical peat fires alone produced an average of 0.1 PgC per year for 1997-
2009, and 0.7 PgC in 1997 in particular (van der Werf et al., 2010). Peat-lands can be significant in
equatorial Asia but also boreal regions where their combustion leads to the release of long-stored carbon
(Turetsky et al., 2015). In 1998 and 2001, the difference in emissions could not be attributed to a
particular location. While fire emissions from Equatorial Asia were underestimated, GFEDv3 observed
lower emissions over Africa compared to INFERNO, which seems to be the key driver of our
discrepancies.
**Table 2. Mean yearly emission budgets in Peta-grams of emitted carbon and mean yearly burnt area budgets**
**in Mkm2 for the 1997-2010 period. Latitudes were bound to: beyond 50º (high latitudes), 35º to 50º (mid-**



latitudes), 15° to 35° (low latitudes) and below 15° (equatorial). Four configurations of INFERNO are
presented, with CRU-NCEP and WFDEI driving meteorology coupled with three ignition modes: mode 1
indicates constant anthropogenic and lightning ignitions, mode 2 is for constant anthropogenic with
interactive lightning ignitions, and mode 3 for interactive lightning and anthropogenic ignitions.

| Emitted carbon (PgC/year) | mode 1 CRU-NCEP | mode 1 WFDEI | mode 2 CRU-NCEP | mode 3 CRU-NCEP |
|---|---|---|---|---|
| **High latitudes** | 0.087 | 0.096 | 0.082 | 0.091 |
| **Mid-latitudes** | 0.185 | 0.193 | 0.170 | 0.191 |
| **Low latitudes** | 0.716 | 0.624 | 0.627 | 0.591 |
| **Equatorial** | 1.157 | 1.130 | 1.021 | 1.385 |


| Burnt area (Mkm$^2$ / year) | mode 1 CRU-NCEP | mode 1 WFDEI | mode 2 CRU-NCEP | mode 3 CRU-NCEP |
|---|---|---|---|---|
| **High latitudes** | 0.176 | 0.196 | 0.162 | 0.179 |
| **Mid-latitudes** | 0.485 | 0.557 | 0.445 | 0.531 |
| **Low latitudes** | 1.648 | 1.884 | 1.558 | 1.531 |
| **Equatorial** | 1.524 | 1.580 | 1.423 | 1.693 |


Table 2 shows the budgets for four latitudinal bands across the various simulations performed. The
second ignition mode (constant anthropogenic and interactive lightning ignitions at any time and place)
appears to consistently predict lower emissions and burnt area (with the exception of low latitudes).
Furthermore, the main impact of using an ignition model that varies with both natural and anthropogenic
ignitions is a reduction of fires at low (tropical and sub-tropical) latitudes, and an increase in equatorial
regions. Indeed, when compared to constant ignitions (mode 1), interactive ignitions (mode 3) predict
more emissions in forest encroachment regions (noticeably surrounding the Congo and Amazon
rainforests), and less in heavily-populated areas (Nigeria, India). Meanwhile, we observed interactive
lightning ignitions (mode 2) significantly reduced burning in grassland-savannah environments. We link
this to the predominance of cloud-to-ground lightning strikes in wet environment within the LIS-OTD
dataset (e.g. the Congo rainforest, (Christian et al., 2003) and fewer strikes (and ignitions) in the more
flammable grasslands and savannahs. These issues are visible in Fig. B1, which shows difference maps
of the four model configurations, for 1997-2010 mean yearly totals. Equatorial and boreal regions include
peat that leads to large fuel consumption, which is unaccounted for in JULES, suggesting that our model
will inherently underestimate emissions from these regions.
In order to examine whether our flammability can represent fire occurrence, three other fire indices were
diagnosed, namely the McArthur, Nesterov and Canadian fire indices. These indices were obtained
seamlessly during the model runs, therefore utilizing the same meteorological and hydrological driving
variables, and the same vegetation condition. Their predictions were regressed with GFEDv4 1997-2010



annual burnt area (Giglio et al., 2013). This analysis relies on the assumption that fire indices can be used
as a proxy for fire occurrence and spread, and eventually burnt area. Only areas that had been observed
to burn sometime between 1997 and 2010 were sampled; to avoid accounting for high fire indices in
non-vegetated areas such as the Sahara.
Table 3 shows the result of our analysis. Ignitions followed mode 1; in this mode ignitions are constant,
therefore the only variability in burnt area (and performance) is due to INFERNO's flammability scheme.
The McArthur index performs poorly at high latitudes (it was made for Australia), but outperforms the
other indices in low latitude regions. The Canadian and Nesterov indices correlate best with observed
burnt area in high latitude regions (for which they were developed). Altogether, INFERNO's burnt area
appears to follow observed burnt area better than the sole usage of a fire index.
**Table 3. Temporal correlation coefficients (R) of annual means (1997-2010) shown for four latitudinal bands.**
**R-coefficients were obtained between either of the three simulated fire indices or INFERNO's burnt area**
**(ubiquitous ignitions – ignition mode 1, using CRU-NCEP meteorology) and burnt area from GFEDv4 (Giglio**
**et al., 2013). We restrict our analysis to grid-boxes in which GFEDv4 observed burning. Latitudes were bound**
**to: beyond 50° (high latitudes), 35° to 50° (mid-latitudes), 15° to 35° (low latitudes) and below 15° (equatorial).**

| R-coefficient (with GFEDv4 burnt area) | INFERNO Burnt area | Nesterov Index | McArthur Index | Canadian Index |
|---|---|---|---|---|
| **Global** | 0.649 | 0.088 | -0.009 | 0.266 |
| **High latitudes** | 0.476 | 0.522 | -0.005 | 0.519 |
| **Mid-latitudes** | 0.179 | -0.006 | 0.069 | 0.060 |
| **Low latitudes** | 0.603 | 0.476 | 0.499 | 0.480 |
| **Equatorial** | 0.689 | 0.239 | 0.354 | 0.392 |


## 345   5 Conclusion

Through a minimalistic approach we propose a parameterization for fire occurrence of appropriate
complexity for application at large spatial scales within an ESM context: the INteractive Fire and
Emission algoRithm for Natural envirOnments (INFERNO). It directly only varies according to
precipitation (and resulting soil moisture), temperature and humidity, and indirectly it utilizes vegetation.
It is also capable of explicitly simulating ignitions using lightning and anthropogenic information. While
our scheme manages to represent fire occurrence on large scales (both spatial and temporal), it performs
best at low latitudes. INFERNO's burnt area scheme appears superior to the use of fire indices alone
(Nesterov, McArthur and basic Canadian) for capturing annual burnt area variations, and thus one form
of fire impact. However, due to the nature of our analysis (fire danger and burnt area remain different
quantities) this does not imply INFERNO should supersede fire weather indices for operational purposes,
neither has our algorithm been built for numerical weather prediction or seasonal fire danger forecasting.
Nonetheless, our current simulations suggest the variability in emissions is underestimated by
INFERNO, in particular the impact of the 1997-1998 El-Niño and the subsequent La Niña, which may



be attributable to the lack of representation of peat in the model, critical to biomass burning in equatorial Asia and boreal areas. The use of different present-day meteorological datasets, in particular precipitation, has an important impact on the magnitude and variability of our diagnostics. Using WFDEI-GPCC rather than CRU-NCEP led to more burnt area but lower fuel consumption and eventually less emitted carbon (this follows from grasslands burning rather than forests). Vegetation zone interfaces were key to this difference. Similarly, lightning appears to ignite more fires in wet environments (rainforests) while flammable environments (savannah, grasslands) are sensitive to the presence of an ignition source. Including a scheme to parameterise human impacts appears to significantly reduce fires in heavily populated areas, while favouring their encroachment of rainforests (the vicinity of which are an anthropogenic ignition 'sweet spot' in our parameterization). Nevertheless there is much uncertainty attributed to human induced emissions and effects on fire regime (Marlon et al., 2008; Thonicke et al., 2010). Accordingly, we include different modes of ignition to dampen the impact of this uncertainty in INFERNO.

The implementation of INFERNO within the Met Office's Unified Model and its significance for present-day atmospheric composition and climate will be investigated in a separate paper. While a strength of the model is its minimalistic approach the scheme holds potential for improvements: while litter influences flammability, only live vegetation is vaporized. In reality, litter is observed to burn more than live vegetation. Similarly, we predict that the inclusion of peat within JULES would improve its fire diagnostics, especially for locations with large fuel consumptions (e.g. equatorial Asia and boreal climates; van der Werf et al., 2010). Given the predictability of emissions from peat fires in relation with precipitation (van der Werf et al., 2008), this would be a promising area of exploration. The value of this model being its simplicity and linearity, any improvements to INFERNO's meteorological and hydrological assimilation need to remain minimalistic; complex parameterizations are better suited for more specialized fire schemes (Lasslop et al., 2014; Li et al., 2013, p.1).

**Code availability**

Information on the JULES land surface model can be found at: http://jules-lsm.github.io/. INFERNO is included in JULES vn4.5 and is included in this documentation. The JULES source code can be accessed via the Met Office's science repository (requires registration): https://code.metoffice.gov.uk/trac/jules. In particular, the version of the code used to produce the outputs included in this study can be accessed at:
https://code.metoffice.gov.uk/trac/jules/browser/main/branches/dev/stephanemangeon/vn4.3.1_inferno.

**Appendix A**

This appendix contains additional information relating to the INFERNO scheme.





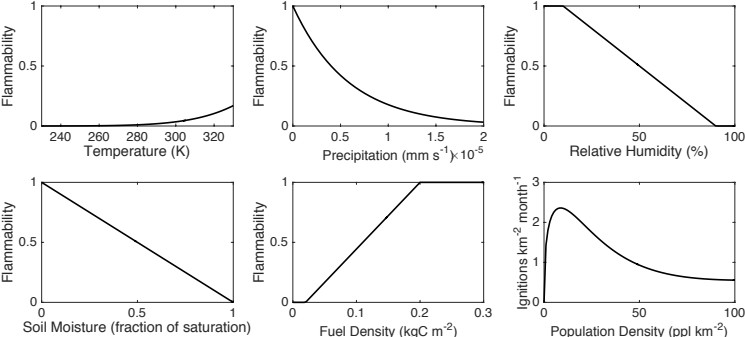


**Fig. A1. The individual dependencies of INFERNO on key driving variables. Note the population density only**
**influences the model output if ignition mode 3 is selected (interactive lightning and human ignition).**
**Table A1. The key JULES PFT-specific parameters for allometry and vegetation carbon used in our**
**simulations (Clark et al., 2011).**

| | Specific leaf density $\sigma_l$ (kg C m$^{-2}$) | Allometric coefficient $a_{wl}$ (kg C m$^{-2}$) | Allometric exponent $b_{wl}$ | Associated Fire Biome in Akagi et al., 2011 |
|---|---|---|---|---|
| **Broadleaf Evergreen Tree (Tropical)** | 0.0375 | 0.65 | 1.667 | Tropical Forests |
| **Broadleaf Evergreen Tree (Temperate)** | 0.0375 | 0.65 | 1.667 | Temperate Forests |
| **Broadleaf Deciduous Tree** | 0.0375 | 0.65 | 1.667 | Tropical Forests |
| **Needleleaf Evergreen Tree** | 0.1 | 0.65 | 1.667 | Temperate Forests |
| **Needleleaf Deciduous Tree** | 0.1 | 0.75 | 1.667 | Boreal Forests |
| **C3 grass** | 0.025 | 0.005 | 1.667 | Temperate Forests |
| **C4 grass** | 0.05 | 0.005 | 1.667 | Savannah and Grasslands |
| **Evergreen Shrub** | 0.05 | 0.10 | 1.667 | Temperate Forests |
| **Deciduous Shrub** | 0.05 | 0.10 | 1.667 | Boreal Forests |


**Table A2. The characteristics of the Canadian's Fire Weather Index's three fuel moisture codes.**

| | Type of fuel | Dry weight (kg m$^{-2}$) | Time lag (days) | Water capacity (mm) |
|---|---|---|---|---|
| **Fine Fuel Moisture Code** | Litter and other fine fuels | 0.25 | 2-3 | 0.6 |
| **Duff Moisture Code** | Loosely compacted decomposing organic matter | 5 | 12 | 15 |





| | | | | | |
|---|---|---|---|---|---|
| **Drought Code** | Deep layer of compact organic matter | | 25 | 52 | 100 |

**Appendix B**

This appendix contains additional results illustrating the dependence of INFERNO with ignitions and its performance on a regional basis.

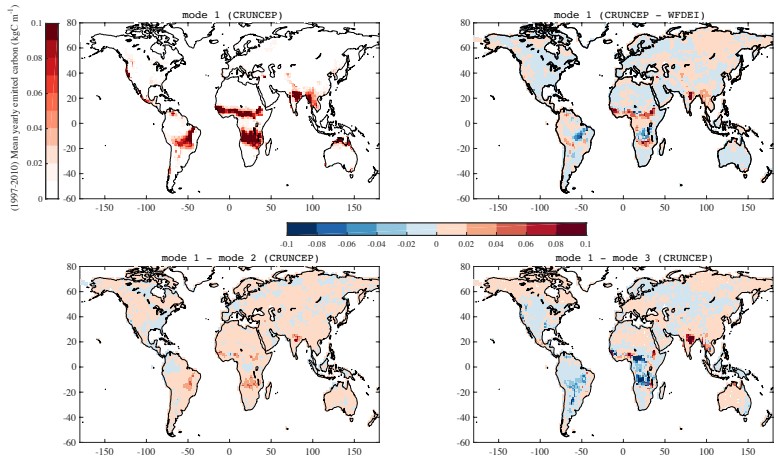

**Fig. B1. Emitted carbon difference maps between the four runs performed to analyse the sensitivity of INFERNO to ignitions (our three ignition modes, see Sect. 2.1.1) and meteorology (CRUNCEP and WFDEI-GPCC).**

**Table B1. Regional budgets according to the standard GFED regions (van der Werf et al., 2010).**

| GFED standard regions | Mean Yearly Burnt Area (in Mha) | | Mean Yearly Emitted Carbon (in TgC) | |
|---|---|---|---|---|
| | GFED4* | INFERNO** | GFED3*** | INFERNO** |
| **Boreal North America** | 2.2 | 5.2 | 54 | 37 |
| **Temperate North America** | 1.8 | 29.9 | 9 | 106 |
| **Central America** | 1.8 | 7.9 | 20 | 45 |
| **Northern Hemisphere South America** | 2.6 | 4.0 | 22 | 51 |
| **Southern Hemisphere South America** | 18.7 | 68.3 | 271 | 483 |
| **Europe** | 0.7 | 5.0 | 4 | 29 |
| **Middle East** | 0.8 | 12.3 | 2 | 19 |





| | | | | |
|---|---|---|---|---|
| **Northern Hemisphere Africa** | 117.7 | 120.4 | 481 | 533 |
| **Southern Hemisphere Africa** | 125.0 | 57.6 | 557 | 610 |
| **Boreal Asia** | 5.6 | 9.7 | 128 | 55 |
| **Central Asia** | 13.6 | 23.8 | 36 | 50 |
| **Southeast Asia** | 7.0 | 29.6 | 103 | 170 |
| **Equatorial Asia** | 1.6 | 0.5 | 191 | 10 |
| **Australia and New Zealand** | 50.2 | 30.2 | 135 | 96 |

407    * GFED4 mean yearly burnt area from Giglio et al. (2013), from 1997 to 2011. ** INFERNO mean yearly burnt area from 1997

408    to 2010, using ignition mode 3 (varying anthropogenic and natural ignitions) and CRU-NCEP driving meteorology. *** GFED3

409    mean yearly emitted carbon from van der Werf et al. (2010) from 1997 to 2009.





## Author contribution

Apostolos Voulgarakis supervised the scientific design of INFERNO and the writing of this article. Gerd Folberth also supervised these aspects, with an emphasis on technical aspects of INFERNO in relation with the Met Office's Unified Model. Richard Gilham contributed to the technical design of the model and its implementation and led the writing on fire indices. Anna Harper contributed to the design of INFERNO in relation to the vegetation scheme's recent development, helped with the analysis of vegetation biases in the study's results and led the writing on the vegetation scheme. Stephen Sitch contributed throughout the writing, analysis and the scientific design of this study.

## Acknowledgements

We wish to thank Robert Field, Pierre Friedlingstein, Stephen Hardwick, Sandy Harrison, Colin Prentice, Eddie Robertson and Andy Wiltshire for their inputs in the development and design of INFERNO; Olga Pechony, Greg Faluvegi and Drew Shindell for sharing their work on a fire parameterization. The lead author gracefully thanks the Natural Environment Research Council (NERC, UK) and the UK Met Office for ongoing financial support, as well as the European Commission's Marie Curie Actions International Research Staff Exchange Scheme (IRSES) for past support under the REQUA project.

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
