# Peer review of "INFERNO: a fire and emissions scheme for the UK Met"

_Geoscientific Model Development, 2016_

## Referee Comment (RC1) · Anonymous Referee #1 · 23 Mar 2016

Mangeon et al. present the new fire and emissions schem ofr the Met Office's unified model. The approach presented has a reasonable complexity for to be useful in an Earth system model. The model is evaluated using two different forcing datasets and different configurations of the ignition parameterization. Additionally the model performance is compared to the performance of fire weather indices. Overall this is an interesting presentation suitable for publication within GMD. Nevertheless I have a number of suggestions which I believe will help to strengthen and improve the manuscript.

General comments:

The comparison with GFED focusses on stating that the emissions due to peat fires cannot be reproduced by a model not including peatlands. This is correct, a solution could be to exclude the emissions from peatlands from the comparison, as GFED

provides the emissions for a number of different sources.

I find the term fuel density to describe the amount of fuel per m-2 a bit confusing, as this term is often used (for instance within spitfire) as the amount of fuel per volume. If it is the density per volume then the rate of spread decreases with increasing density. I would prefer the term fuel load.

A paragraph specifying the datasets used for the model evaluation is missing. The evaluation could also be a bit extended, for instance showing not only results of carbon emissions but also for the different chemical species.

It remains unclear to me whether the fire model affects the vegetation dynamics, is there any tree mortality computed? also whether vegetation dynamics are included in the model simulations. If fire and vegetation dynamics interact a comparison of tree cover would be useful to evaluate that part of the model. If not, why don't they?

Specific comments: l. 19: you could add the outcome of the fire index diagnostics comparison.

l. 21: is this spatial or temporal correlation? Is it significant?

l. 101: the scaling factor is the 7.7, please specify.

l. 102-5: if you assume fNS=1, you don't need it in the equation, adding this assumption after presenting the equation might be more clear: total ignitions can be represented as: eq3, here fNS equals 1 for mode 1 and 3 and follows eq. 2 for mode 3.

l. 117: Leaf carbon is the living biomass?

l. 119: I think this should say $FD_{PFT}$, the equation actually does not scale lienarly between 0 and 1, it jumps from 0 to $Fuel_{low}/(Fuel_{high} - Fuel_{low})$. I guess the equation should be $((FPMc + leafC) - Fuel_{low})/(Fuel_{high} - Fuel_{low})$. Additionally, the equation is not defined for fuel density being equal to fuellow and fuelhigh.

l. 123: Here again, the normalization term should be $(RH - RH_{low})/(RH_{up} - RH_{low})$,

please rewrite the equation similar to eq.5 to define the bounds for relative humidity being higher and lower then the thresholds.

l. 127: FD is Fuel Density or Fuel density index?

l. 126-127: which formula are you referring to?

l. 133: explain how the average burnt area was determined. There is no difference between temperate and tropical trees?

l. 146: That suggests you should vary your pft specific burned area. Are there any indications in your results that this is necessary? it might be rather a point for the discussion of your results.

l. 154: CCmin and CCmax are the same for leaves and stems.

l. 158: I don't see why this is makes it justifyable. more interesting would be why you changed the value, was it to tune the emissions?

l. 200: what happens with population and lightning flash rates if JULES is not used in standalone version.

l. 206-225: Give equations for the fire weather indices.

l. 254: How is the correlation computed? spatial or temporal? if temporal, is the correlation computed for each grid cell or just for the global total? pleave give significance levels.

l. 258-262: I don't think the gridcell with maximum burned area is an important benchmark. But what about seasonality? Emissions for the different sources given by GFED could also be interesting. or burned area separated for grass and woody pfts.

l. 264: the peat emissions given by GFED could be excluded here, are crop fires and emissions due to deforestation actually somehow included in the model? otherwise they could also be excluded.

l. 338: interestingly the mid latitudes are not well captured by the fire weather indices. might be the human influence? Including the other ignition modes of the model could give an indication why the model in better than the indices. Any significance values on the correlation?

l. 361: where did you show that the precipitation has an important impact?

l.370: You assessed the uncertainty of the ignitions by including the different ignition modes, but how does this dampen the impact of this uncertainty in inferno?

l. 375: what do you mean by vaporized?

l. 379-382: I don't understand. what do you mean by INFERNO's meteorological and hydrological assimilation? In what sense are the other fire schemes more specialized?

Fig.A1: label the subpanels. why does the temperature function not scale between 0 and 1?

---

## Referee Comment (RC2) · Anonymous Referee #2 · 8 May 2016

Review
INFERNO: A fire and emissions scheme for the Met Office's Unified Model
Mangeon et al.

This paper describes a simplified model that projects biomass burning activity, burnt area, and emissions globally. The model framework uses climatic and meteorological inputs and land cover characteristics to drive the emissions model. The ignition sources can be varied, and, for the purposes of the model evaluation presented, are prescribed three different ways in order to assess the sensitivity of the model to different ignition parameterizations. The land cover inputs are provided by the JULES model. The fire model is run for current conditions and compared with other fire model outputs (primarily GFED).

This paper is written extremely well, and the modeling tool described is a unique contribution. It uses different approaches than other available models that project fires in global climate models and will be a useful tool to be incorporated within the UK Met Office's Earth System Model. The assumptions made in the model parameterizations are reasonable and well justified throughout. The manuscript is very appropriate for *Geoscientific Model Development*, and I recommend publication after only minor comments that I provide here.

General Comments:

Section 2.2: How are the PFTs allocated within each grid cell? This is not explained thoroughly in Section 2.2, and the paper cited as a reference is still "*in prep".*

Section 2.3: Emissions from Akagi et al. (2011) have been updated and can be incorporated within future versions (see Section 3 at http://bai.acom.ucar.edu/Data/fire/).

Section 4: I was confused about the fact that there were two different versions of "GFEDv4" used to evaluate the INFERNO estimates. Can this be made more clearly? (i.e., better define and label the two different outputs)?
Lines 255 and following sentences: Please clarify which model (INFERNO v GFED) was higher/lower. For example, Line 255 can be re-written: "We notice that the burnt area predicted by INFERNO is higher in all regions other than Australia and New Zealand, and southern hemisphere Africa when compared to GFED4."

Paragraph starting at line 325: Is it possible to compare the fire indices calculated here with the real data for current conditions?

Figure 3: Why are not GFAS and FINN outputs compared in both panels of the figure?

*Editorial Comments:*

Title: The UK Met Office should be defined in the title.
Lines 59 and 60: The present tense should be applied (i.e., change "used" to "uses")
Line 67: Add a comma after ($E_x$)
Line 168: It may be useful to let the reader know that [C] will be described in the next section.
Line 180 (and elsewhere): When "which" is used, there should be a comma preceding it. In this case, there should be a comma after "(see Eq. 6.8)"
Line 182: Layers should be plural

Line 258: I recommend changing "observes" to "projects" (or something like that).
Line 328: condition should be plural
Line 365-6: Should this be "…presence of an *anthropogenic* ignition source."
Line 375-376: a citation should be given.

---

## Author Comment (AC1) · 26 Jun 2016

**Response to Reviewers Comment.**

**Reviewer 1 –**

Mangeon et al. present the new fire and emissions schem ofr the Met Office's unified model. The approach presented has a reasonable complexity for to be useful in an Earth system model. The model is evaluated using two different forcing datasets and different configurations of the ignition parameterization. Additionally the model performance is compared to the performance of fire weather indices. Overall this is an interesting presentation suitable for publication within GMD. Nevertheless I have a number of suggestions which I believe will help to strengthen and improve the manuscript.

General comments:

The comparison with GFED focusses on stating that the emissions due to peat fires cannot be reproduced by a model not including peatlands. This is correct, a solution could be to exclude the emissions from peatlands from the comparison, as GFED provides the emissions for a number of different sources.

Reply: This is a fair point highlighted by the reviewer. Nevertheless, we do mention the figures from GFED and have extended on that sentence.
Changes to manuscript: l342-345.Furthermore, 2002 and 2006 also saw important peat burning, with GFEDv3 estimating peat emissions of 0.16 and 0.21 PgC respectively. In both of these years, the trend in INFERNO differs from GFEDv3's (stagnation in 2002 and decrease in 2006).

I find the term fuel density to describe the amount of fuel per m-2 a bit confusing, as this term is often used (for instance within spitfire) as the amount of fuel per volume. If it is the density per volume then the rate of spread decreases with increasing density. I would prefer the term fuel load.

Reply: We welcome the suggestion by the reviewer.
Changes in manuscript: Across the manuscript, fuel density has been replaced with fuel load, and FD with FL.

A paragraph specifying the datasets used for the model evaluation is missing.

Reply: This paragraph was added to the manuscript at the end of the model configuration section.
Changes in manuscript: l272-287. Evaluation was performed against the published data for GFEDv3, FINNv1, GFAS and GFEDv4. We also used the data from GFEDv4s (http://globalfiredata.org, manuscript in preparation) and GFEDv4 (Giglio et al., 2013) to calculate grid-specific emissions and burnt-area. The Global Fire Emissions Database (GFED) passes satellite observation of burnt area through the Carnegie-Ames-Stanford-Approach (CASA) biogeochemical model in order to obtain emissions from open burning. GFEDv4 (Giglio et al., 2013) innovates on GFEDv3 (Giglio et al., 2010)

mainly through an updated algorithm to retrieve burnt area from MODIS satellite products and an increased spatial and temporal resolution, to $0.25^{o}$ and daily (this resolution was assessed in Mangeon et al., 2015). Meanwhile GFEDv4s also includes the contribution from small fires (Randerson et al., 2012).

The Fire Inventory from NCAR version 1.0 (FINNv1, Wiedinmyer et al., 2011) provides high-resolution (both temporal and spatial) global emissions of trace gas and particle emissions from open burning of biomass. It focuses on rapid availability and assimilation in real time forecast and follows a similar process to GFED to estimate emission, but its burnt area is obtained directly from fire pixel using land cover (Wiedinmyer et al., 2011). The Global Fire Assimilation System (GFAS, Kaiser et al., 2012), unlike the aforementioned products, directly assess emissions from satellite-observed fire radiative power more apt at detecting small fires and avoiding the uncertainty of biogeochemical models.

The evaluation could also be a bit extended, for instance showing not only results of carbon emissions but also for the different chemical species.

Reply: We have added total emissions for the different chemical species in Table 3. Still, we plan to further study specific species when performing the atmospheric composition evaluation of INFERNO, when coupled to the atmosphere. Further analysis will be included therein. Regarding the length of the evaluation, it has been increased following specific suggestions from the reviewers.

Changes to manuscript: l373-381. Species-specific average emissions produced by the INFERNO

scheme are shown in Table 3 in Tg per year for the 1997-2010 period. CO and CH4 appear to be produced in noticeably larger quantities than in observation-based emission estimates. This hints at an overrepresentation of smouldering-type combustion. In INFERNO this might be due to the emission factors used, or the type of vegetation burnt. Table 3. Average annual emission (Tg per year) for

INFERNO with the interactive ignition mode and CRUNCEP reanalysis (3 – CRUNCEP) and the constant ignition mode and WFDEI reanalysis (1 – WFDEI), comparison to GFASv1 (Kaiser et al.,

2012), GFEDv3 (van der Werf et al., 2010) and FINNv1 (Wiedinmyer et al., 2011) is provided.

It remains unclear to me whether the fire model affects the vegetation dynamics, is there any tree mortality computed? also whether vegetation dynamics are included in the model simulations. If fire and vegetation dynamics interact a comparison of tree cover would be useful to evaluate that part of the model. If not, why don't they?

Reply: At this stage, we focused on providing diagnostic tools with INFERNO. Having a vegetation that interacts with fires (carbon removal and tree mortality) might be investigated in the future. This would require a much deeper investigation and calibration of vegetation within JULES, beyond the scope of our work.

Changes in manuscript: l215-217. In its current early state, INFERNO provides a diagnostic tool, it does not remove carbon from vegetation nor does it lead to tree mortality. l431-434. To close the vegetation- fire feedback, INFERNO will eventually need to remove carbon from vegetation and to include tree
mortality.
Specific comments:
l. 19: you could add the outcome of the fire index diagnostics comparison.
Reply: Thanks
Changes to manuscript: l19-20. We found INFERNO captured global burnt area variability better than
individual indices, and these performed best for their native regions.
l. 21: is this spatial or temporal correlation? Is it significant?
Reply: This is spatial correlation. We have modified the latter text (to keep the abstract short).
Changes to manuscript: l293-294. we found a spatial correlation of R=0.66 for burnt area and R=0.59
for emitted carbon, both passing the t-test with 95% significance.
l. 101: the scaling factor is the 7.7, please specify.
Reply: This is now specified in the text.
Changes in manuscript: l. 102. Equation 2 includes a scaling factor of 7.7 ([…]
l. 102-5: if you assume fNS=1, you don't need it in the equation, adding this assumption after presenting
the equation might be more clear: total ignitions can be represented as: eq3, here fNS equals 1 for mode
1 and 3 and follows eq. 2 for mode 3.
Reply: Thanks for the comment, we clarified it as suggested in the manuscript.
Changes to the manuscript: l103-107. Total ignitions […] Here fNS = 1 for mode 1 and 2 and follows
eq. 2 for mode 3.
l. 117: Leaf carbon is the living biomass?
Reply: Yes – this was clarified as aboveground to only represent living leaves
Changes in the manuscript: l119. (LeafC,PFT,  aboveground)
l. 119: I think this should say FD$_{\text{PFT}}$, the equation actually does not scale lienarly between 0 and 1, it
jumps from 0 to $\text{Fuel}_{\text{low}}/(\text{Fuel}_{\text{high}} -\text{Fuel}_{\text{low}})$. I guess the equation should be
$((\text{FPMc}+\text{leafC})-\text{Fuel}_{\text{low}})/(\text{Fuel}_{\text{high}}-\text{Fuel}_{\text{low}})$. Additionally, the equation is not defined for fuel density
being equal to fuellow and fuelhigh.

Reply: A great thanks to the reviewer for spotting this, the equation was changed to reflect this.

Changes in the manuscript: l118. Eq. 5.

$$FL_{PFT} = \begin{cases} 1 \text{ for } Fuel_{high} < (DPM_C + Leaf_{C,PFT}) \\ \dfrac{(DPM_C + Leaf_{C,PFT}) - Fuel_{low}}{Fuel_{high} - Fuel_{low}} \text{ for } Fuel_{low} \leq (DPM_C + Leaf_{C,PFT}) \leq Fuel_{high} \\ 0 \text{ for } Fuel_{low} > (DPM_C + Leaf_{C,PFT}) \end{cases}$$

l. 123: Here again, the normalization term should be $(RH - RH_{low})/(RH_{up} - RH_{low})$, C2 please rewrite the equation similar to eq.5 to define the bounds for relative humidity being higher and lower then the thresholds.

Reply: Again, thanks to the reviewer for spotting this and suggesting the edit

Changes in the manuscript: l125.

$$F_{PFT} = \begin{cases} e^* \dfrac{RH - RH_{low}}{RH_{up} - RH_{low}} e^{-2R} FL_{PFT} (1 - \theta) \quad \begin{array}{c} e^* e^{-2R} FL_{PFT} (1 - \theta) \text{ for } RH_{up} < RH \\ \text{for } RH_{low} \leq RH \leq RH_{high} \\ 0 \text{ for } RH_{low} > RH \end{array} \end{cases}$$

l. 127: FD is Fuel Density or Fuel density index? l. 126-127: which formula are you referring to?

Reply: FD (or now FL) stands for fuel load index here (this was clarified). We also clarified the term further to avoid confusion.

Changes to the manuscript: l129. fuel load (DPMc and Leafc,pft). (l. 153) Fuel Loaf index (FL)

l. 133: explain how the average burnt area was determined. There is no difference between temperate and tropical trees?

Reply: We specified the average burnt area was *heuristically* determined. Far from stating this method is perfect, we feel within this paragraph and the next we give enough justification for it while also presenting the alternatives. Temperate and tropical trees are assigned the same average burnt area, for simplicity.

Changes to the manuscript: l137. Sub-categories of trees, grass and shrubs are not differentiated

l. 146: That suggests you should vary your pft specific burned area. Are there any indications in your results that this is necessary? it might be rather a point for the discussion of your results.

Reply: This is difficult to ascertain without comparing earlier components (fire count or frequency).

Which we have not done for this paper (and INFERNO does not have the capability to produce fire count estimates). Thank you to the reviewer for suggesting the discussion was a more appropriate location for this point, which we have also extended upon.

Changes to manuscript: l307-311. INFERNO's approach to burnt area only considers trees, grass and shrub cover and was determined heuristically, meanwhile Hantson et al. (2014) found global fire size was mostly influenced by precipitation, aridity and human activity (population density and croplands).
Further parameterizations for fire size exist (e.g., Hantson et al., 2015, 2016) which could improve
INFERNO burnt area estimates while maintaining simplicity and traceability.
l. 154: CCmin and CCmax are the same for leaves and stems.
Reply: Again, thanks to the reviewer for spotting this. CC (stem) was modified accordingly.
Changes to the manuscript: l157. […] and stems (CCmin = 0.0 and CCmax = 0.4) […]
l. 158: I don't see why this is makes it justifyable. more interesting would be why you changed the value,
was it to tune the emissions?
Reply: We realize stating 'justifiable' implies GFED's assumptions should be mimicked. We have
rephrased it to convey the message more effectively. We also go into more depth on the matter.
Changes to the manuscript: l160-164. Nevertheless, GFED uses a more complex representation of
moisture across multiple fuel types and only accounts for fires that were observed. In comparison, our
scheme only relies on soil moisture and was much more sensitive to minimum combustion, such that the
contribution from moist forested areas (e.g., rainforests) needed to be reduced by increasing the impact
of soil moisture (reducing stems' $CC_{min}$).
l. 200: what happens with population and lightning flash rates if JULES is not used in standalone version.
Reply: Ignition mode 2 is essentially the coupled version (to be refined and later submitted for
publication): the lightning can be interactively simulated (population still needs to be prescribed or
assumed as constant).
Changes to the manuscript: l211-212. Interestingly, lightning can be interactively simulated in
atmospheric models (not population), although this will not be explored in this paper.
l. 206-225: Give equations for the fire weather indices.
Reply: We have added equations for the more straightforward McArthur and Nesterov index. For the
Canadian FWI we have only included the key equation, the full index is very complex and we refer back
to the original papers for a detailed description.
Changes to manuscript: l226-250. Equation 12-15 and related details in section 2.3. Appendix A now
contains the equations used in the Canadian Fire Weather Index.
l. 254: How is the correlation computed? spatial or temporal? if temporal, is the corre- lation computed
for each grid cell or just for the global total? pleave give significance levels.

Reply: Similar to previous point. This is spatial correlation, which is computed for each gridcell. We
found p-values of 0 for both correlations (and these passed the t-test with 95% significance level). There
are about 28000 grid-cells included in the analysis which explains the virtually 0 p-value here.
Changes to manuscript: l292-296. INFERNO accurately diagnoses total fire occurrence and
emissions over the 1997-2010 period: we found a spatial correlation of R=0.66 for burnt area and R=0.59
for emitted carbon, both passing the t-test with 95% significance.
l. 258-262: I don't think the gridcell with maximum burned area is an important bench- mark. But what
about seasonality? Emissions for the different sources given by GFED could also be interesting. or
burned area separated for grass and woody pfts.
Reply: We wrote the maximum burnt area for two reasons: 1. It mirrors the emitted carbon, where the
maximum is much more indicative of a model bias (with peat). 2. It remains important to assess whether
INFERNO can assess extreme burnt area, and it adds a narrative to the article. Therefore, we would
prefer to keep this point, while expanding others as suggested. Nevertheless, we will follow the
suggestions of the reviewer and mention the main PFTs that contribute to burnt area, and emissions (C4
grass and Broadleaf Evergreen Tree (Tropical)). More details were also added for peat-specific GFED
emissions (see previous comment response).
Changes to manuscript: l299-301. Over the studied period, C4 grass were the main contributors to burnt
area in INFERNO (a mean 2.34 Mkm$^2$ per year), meanwhile Broadleaf Evergreen Trees (Tropical) led
to the most emitted carbon (a mean 1.48 Pg per year).
l. 264: the peat emissions given by GFED could be excluded here, are crop fires and emissions due to
deforestation actually somehow included in the model? otherwise they could also be excluded.
Reply: Although these are not explicitly represented in the model, the function that estimates ignitions
according to population density represents the various ways humans can interact with fires – including
deforestation and crop fires (although in these simulations, crops are equivalent to grass). Peat and
crop/deforestation fire emissions are different in that the latter are somehow accounted for (albeit not
explicitly), while the former simply is not present in the land surface model, and thus in INFERNO. We
detail the contribution of peat fires to GFED later in the discussion, and have expanded on that analysis.
Changes to manuscript: l342-345. Furthermore, 2002 and 2006 also saw important peat burning, with
GFEDv3 estimating peat emissions of 0.16 and 0.21 PgC respectively. In both of these years, the trend
in INFERNO differs from GFEDv3's (stagnation in 2002 and decrease in 2006).
l. 338: interestingly the mid latitudes are not well captured by the fire weather indices. might be the
human influence? Including the other ignition modes of the model could give an indication why the
model in better than the indices. Any significance values on the correlation?

Reply: Thanks the reviewer for this comment – we performed further analysis with ignition mode 3 and
found a correlation coefficient of R=0.1617 (lower than mode 1), suggesting our scheme for interactive
human ignition is not able to improve estimates at mid-latitudes. We also performed a t-test (95%
significance) on the values for correlations, and they were all passed successfully but for global and high
latitudes for the McArthur index, and mid-latitude for the Nesterov index.
Changes to manuscript: Italics mean the correlation was not significant (p-value above 0.05).
l. 361: where did you show that the precipitation has an important impact?
Reply: During the analysis of our results we found precipitation varied significantly between the two
driving datasets (CRUNCEP and WFDEI-GPCC). Still, we do not wish to do a full analysis of the causes
for this discrepancy and its impact on INFERNO here, and chose to simply remove this confusing term
(indeed Fig. 3 and Table 2 both show the impact of using a different meteorological dataset).
Changes to manuscript: l418. The use of different present-day meteorological datasets has an important
impact on the magnitude and variability of our diagnostics.
l.370: You assessed the uncertainty of the ignitions by including the different ignition modes, but how
does this dampen the impact of this uncertainty in inferno?
Reply: Thank you for the comment – the language was poorly chosen, and the purpose of these multiple
ignition is to assess rather than dampen indeed.
Changes to manuscript: l428-429. Accordingly, we include different modes to examine the impact of
ignitions (human or natural) in INFERNO.
l. 375: what do you mean by vaporized?
Reply: Thanks to the reviewer for this comment – vaporized was not the most accurate term here, we
modified the sentence accordingly.
Changes to manuscript: l434-436. For instance, litter influences flammability but only live vegetation
leads to emissions while in reality litter significantly contributes to observed fuel consumption (van
Leeuwen et al., 2014).
l. 379-382: I don't understand. what do you mean by INFERNO's meteorological and hydrological
assimilation? In what sense are the other fire schemes more specialized?
Reply: This was referring to the way INFERNO assimilates weather and water. However, the goal of
this sentence is to advise anyone that would develop INFERNO further to keep its 'simplicity and
linearity'. We have changed the sentence to reflect this. What we meant by specialized fire scheme is
more often referred to as 'process-based models'.

Changes to manuscript: l440-441. The value of this model being its simplicity and linearity, any
improvements to INFERNO should follow this vision; complex parameterizations are better suited for
process-based fire schemes (e.g., Lasslop et al., 2014; Li et al., 2013, p.1).
Fig.A1: label the subpanels. why does the temperature function not scale between 0 and 1?
Reply: Regarding the temperature, we chose to restrict the display here to the range of realistic
temperatures observed on Earth (similarly, precipitation and fuel density are restricted to the 'key'
location of their representative functions). Regarding the request to label the subpanels – it's now done.
Changes to manuscript: l452. Updated Fig. A1 and changed its caption to:
Fig. A1. The mathematical functions used for individual dependencies of INFERNO on key driving
variables for flammability (a,b,c,d,e) and ignitions (f), within the range of reasonable earth observations.
Note the population density only influences the model output if ignition mode 3 is selected (interactive
lightning and human ignition).

**Reviewer 2 –**
Review
INFERNO: A fire and emissions scheme for the Met Office's Unified Model
Mangeon et al. This paper describes a simplified model that projects biomass burning activity, burnt
area, and emissions globally. The model framework uses climatic and meteorological inputs and land
cover characteristics to drive the emissions model. The ignition sources can be varied, and, for the
purposes of the model evaluation presented, are prescribed three different ways in order to assess the
sensitivity of the model to different ignition parameterizations. The land cover inputs are provided by
the JULES model. The fire model is run for current conditions and compared with other fire model
outputs (primarily GFED).
This paper is written extremely well, and the modeling tool described is a unique contribution. It uses
different approaches than other available models that project fires in global climate models and will be
a useful tool to be incorporated within the UK Met Office's Earth System Model. The assumptions made
in the model parameterizations are reasonable and well justified throughout. The manuscript is very
appropriate for Geoscientific Model Development, and I recommend publication after only minor
comments that I provide here.
General Comments:
Section 2.2: How are the PFTs allocated within each grid cell? This is not explained thoroughly in Section
2.2, and the paper cited as a reference is still "in prep".

Reply: Given the paper is about fire and there's no feedback on vegetation, we would prefer not to detail
this further. However, we have made this more explicit in the manuscript and the reference (now accepted
in GMD).
Changes in manuscript: l293-195. Fractional coverage of PFTs in any gridcell is based on competition
for resources (light and water), and governed by Lotka-Volterra competition equations, and based on a
tree-shrub-grass dominance hierarchy (Cox, 2001).
Section 2.3: Emissions from Akagi et al. (2011) have been updated and can be incorporated within future
versions (see Section 3 at http://bai.acom.ucar.edu/Data/fire/).
Reply: Now mentioned in the text
Change in the manuscript: l168-169. Section 2.1.5, added: "Updated EFs for Akagi et al. (2011) were
not used in this version of INFERNO, these can be found in section 3 of:
http://bai.acom.ucar.edu/Data/fire/"
Section 4: I was confused about the fact that there were two different versions of "GFEDv4" used to
evaluate the INFERNO estimates. Can this be made more clearly? (i.e., better define and label the two
different outputs)?
Reply: Clarification added in new paragraph on datasets used for comparison (which describes datasets
compared to). Also improved the legend in figure 3 (v4 + small fires).
Changes in manuscript: l272-287 (new paragraph) and Fig. 3 GFEDv4s (v4 + small fires).
Lines 255 and following sentences: Please clarify which model (INFERNO v GFED) was higher/lower.
For example, Line 255 can be re-written:
"We notice that the burnt area predicted by INFERNO is higher in
all regions other than Australia and New Zealand, and southern hemisphere Africa when compared to
GFED4."
Reply: Rephrased this sentence to clarify the comparison
Changes in manuscript: l295-296. Compared to GFEDv4, we notice INFERNO estimates higher burnt
area in all regions apart from Australia and New Zealand, and southern hemisphere Africa.
Paragraph starting at line 325: Is it possible to compare the fire indices calculated here with the real data
for current conditions?
Reply: Fire indices were developed as a means to assess fire dangers, within specific biomes, while they
did not aim to assess 'real data' like burnt area. However, by assessing the R-coefficient with GFED
burnt area we study the variability of the indices, and their capacity to mimic the variability observed in

'real data'. This comparison makes fire indices analogous to INFERNO's flammability, which in ignition
mode 1 (used in this comparison with fire indices), is the only source of variability.
Changes in manuscript: l386-388. This analysis relies on the assumption that fire indices can be used as
a proxy for the variability of fire occurrence and spread, and eventually of burnt area (not the magnitude).
Figure 3: Why are not GFAS and FINN outputs compared in both panels of the figure?
Reply: Neither of the original papers presented results for the other category, i.e. FINN
(http://www.geosci-model-dev.net/4/625/2011/gmd-4-625-2011.html) does not contain information on
Emitted Carbon (although it does on Biomass burned, conversion is not obvious). Meanwhile GFAS
does not estimate burnt area. Accordingly, we also use GFAS, GFEDv3 and FINN to examine specific
species.
Changes to the manuscript: Table 3. & l272. Evaluation was performed against the published data for
GFEDv3, FINNv1, GFAS and GFEDv4.
Editorial Comments:
Title: The UK Met Office should be defined in the title.
Reply: The title now includes UK. However due to branding Met Office should remain so (rather than
Meteorological Office).
Changes to manuscript: title: l1. INFERNO: a fire and emissions scheme for the UK Met Office's Unified
Model.
Lines 59 and 60: The present tense should be applied (i.e., change "used" to "uses")
Reply: OK
Changes to manuscript: l60-62. In short, that parameterization uses monthly mean temperature, relative
humidity and precipitation to simulate fuel flammability. It also uses human population density and
lightning to represent ignitions.
Line 67: Add a comma after (Ex)
Reply: OK
Changes to manuscript: l68 (Ex), and
Line 168: It may be useful to let the reader know that [C] will be described in the next section.
Reply: To an extent we felt the next section did not describe this well enough, therefore we modified this
sentence to be more explicit.
Changes to manuscript: l175-176. and $[C]$ is the dry biomass carbon content which we assume as 50%
(a common simplification; Lamlom and Savidge, 2003).

Line 180 (and elsewhere): When "which" is used, there should be a comma preceding it. In this case,
there should be a comma after "(see Eq. 6.8)"
Reply: Yes, because the clause is restrictive here, "which" should be preceded by a comma.
Changes to manuscript: l187. (see Eq. 6.8), which
Line 182: Layers should be plural
Reply: Yes, thanks
Changes to manuscript: l189. Layers
Line 258: I recommend changing "observes" to "projects" (or something like that).
Reply: Thanks to the reviewer for this suggestion that we applied.
Changes to manuscript: l301. "GFEDv4 projects the" … note: we have also changed all mentions of
GFED4 to GFEDv4 for consistency.
Line 328: condition should be plural
Reply: Yes
Changes to manuscript: l385. same vegetation conditions
Line 365-6: Should this be "...presence of an anthropogenic ignition source."
Reply: The point we were trying to make would apply to both anthropogenic and lightning-started fires.
Albeit it was not clear, which we tried to improve (see changes below).
Changes to manuscript: l422-424. Similarly, lightning appears to more frequently ignite fires in wet
environments (rainforests) while flammable environments (savannah, grasslands) with rarer lightning
are sensitive to the presence of an anthropogenic ignition source.
Line 375-376: a citation should be given.
Reply: The sentence was slightly changed upon reading the study now cited (van Leeuwen et al., 2014)
and to improve its flow.

[revised manuscript text omitted]